# Development and Validation of the Interpersonal Motivational Systems Questionnaire (IMS-Q)

**DOI:** 10.3390/bs13090784

**Published:** 2023-09-21

**Authors:** Rosario Esposito, Stefania Prevete, Concetta Esposito, Dario Bacchini

**Affiliations:** 1School of Cognitive Psychotherapy (SPC), 80122 Naples, Italy; rosaverde@libero.it (R.E.); stefania.prevete@hotmail.it (S.P.); 2Department of Humanities, University of Naples “Federico II”, 80133 Naples, Italy; concetta.esposito3@unina.it

**Keywords:** interpersonal motivational systems, evolutionary framework, AIMIT, questionnaire, validation

## Abstract

Framed within the evolutionary framework, the Interpersonal Motivational System (IMS) theory suggests that eight distinct motivational impulses drive interpersonal human relationships, namely caregiving, social affiliation, attachment, rank-dominance, rank-submission, social play, cooperation, and sexuality. This theory has been widely applied in clinical practice, where psychopathology is viewed as the result of non-flexible or excessive activation of one system over another. Despite its clinical relevance, empirical studies aimed at measuring IMSs are scarce. This paper contributed to filling this gap by proposing a questionnaire to measure individuals’ activation of the eight IMSs. Two studies involving large samples of adults were conducted. The first study (N = 455; 76.5% females) concerned the development of the questionnaire and examination of its content validity through explorative factor analysis. In the second study (N = 635; 54.8% females), confirmatory factor analyses were performed to further refine and confirm the instrument’s factor structure. The final version consisted of 50 items. Empirical validity was established by investigating the correlations between the eight IMSs and other related measures (i.e., personality traits, human basic values, and attachment dimensions). The findings suggest that the IMS framework can be used to understand individual differences in motivation and behavior in different social contexts.

## 1. Introduction

The thesis that human behavior is driven by a limited set of basic motivational factors has an old tradition in the psychological literature [1,2,3]. In recent years, there has been a growing interest in research on motivation, driven by efforts to establish a connection between the evolutionary paradigm and human behavior and motivation.

Most of the more recent theoretical models [4,5,6] are based on the assumption that during evolution, the human brain has become progressively specialized in activating (or deactivating) pre-programmed responses to external or internal triggers [7]. These responses are driven by motivations and allow individuals to adapt optimally to the environment. Human motivation plays a crucial role also in mental health, as psychopathology is often seen as the outcome of unbalanced conflict among motivations or between internal motivation and the external world [6,8,9,10,11,12,13,14,15].

Currently, a unified theory of human motivation is lacking, and efforts to develop valid theories have led to the creation of various measurement instruments. This paper focuses on the Interpersonal Motivational System (IMS) theory developed by Liotti and collaborators [13,16], positing that eight independent motivational impulses guide interpersonal human relationships. The theory, stemming from the evolutionary framework, integrates earlier models of motivational systems model [8,10,11] and has found wide application in clinical practice, where psychopathology is viewed as the non-flexible use or excessive preponderance of one system on another or, in simpler terms, as the consequence of a rigid and dysfunctional activation of the motivational systems associated with the individual’s primary maladaptive interpersonal schemas [17]. The IMS theory seems also a promising approach in the study of personality, as it sheds new light on the complex interplay between traits, emotions, and motivation. However, empirical studies aimed to develop instruments to measure IMSs empirically are relatively scarce [18,19,20,21]. This study contributes to filling a gap in the literature by proposing a new questionnaire to measure the eight IMSs as theorized by Liotti and Monticelli [13].

### 1.1. Evolutionary Basis of Interpersonal Motivational Systems

IMSs are sets of innate rules aimed to ensure individuals’ and specie’s survival [13]. The first theories of motivational systems originated more than 100 years ago in the context of instinct psychology [22]. The contributions of Bowlby [8] and Gilbert [9] marked a significant milestone, paving the way for the current theories of IMS that have been extended and refined by evolutionary psychologists [4,15,23,24]. These systems involve cognitive, emotional, and behavioral components that regulate the relationship between individuals and the environment through cycles of activation and de-activation of different motivational systems based on specific goals. Many theoretical models fall under the umbrella of IMSs, sharing the assumption that they have emerged during the course of human evolution. Darwin’s theory of evolution by natural selection [25] has been influential in shaping current theories on human interpersonal motivation. In particular, Darwin’s ideas about the adaptive function of certain behaviors and traits were first actualized within the ethological foundation of attachment theory [8], positing that attachment is a primary, evolutionarily based motivation that expresses the need for safety and protection in children.

The present paper does not delve into the task of resolving the ongoing debate surrounding aspects like the purpose, operations, quantity, and mechanisms of motivational systems. Instead, our focus lies on the proposal advanced by Liotti and colleagues [13,16], who developed a comprehensive theory of IMSs based on the above-described theoretical background. The theory drew significant inspiration from Gilbert’s model [10,26], which postulated four motivational systems defined as social mentalities and corresponding to care-seeking, caregiving, cooperation, and competition. Gilbert’s model had the merit to have highlighted the biological foundation of human cooperation, a topic that has received increasing attention from many other theorists in recent years [27]. Highly relevant was also the contribution by Lichtenberg to the clinical field [11], who sought to unify motivational theories with the psychoanalytic theory, postulating the existence of seven motivational systems: physiological regulation, attachment, exploration and assertion of preferences, aversiveness, sensuality/sexuality, affiliation with a group, and caregiving. 

In Liotti’s approach, human functioning can be explained by referring to a wide array of motivational systems whose function is to allow individuals to achieve biosocial goals of high evolutionary value, adaptive for the individual, the social group, and the species [16,28]. The eight IMSs are labeled as follows:
(1) the attachment system aims to secure the protection of an attachment figure when one perceives oneself as vulnerable. In Bowlby’s theory [8], attachment motivation is triggered by the need for protection and safety when feelings of fear of abandonment and insecurity arise;(2) the caregiving system aims to protect another conspecific perceived as vulnerable [29]. It involves the feeling of sympathy, concern, or protective tenderness toward the other’s needs;(3–4) the Ranking system, which takes two forms of dominance and submission, aims to maintain the hierarchical organization of the group [9,10]. These systems were shaped by our long primate history of hierarchical social interactions [30], stemming from the limited access to resources in nature which enhance virtues of leadership and followership, dominant vs. submitted;(5) the sexual system aims to form a stable couple’s relationship. It refers to achieving sexual pleasure and reproductive success through sexual interaction [31];(6) the cooperation system aims to achieve a common goal [32]. It refers to the activation of mutual and socio-reciprocal behaviors based on equality and symmetry among the individuals involved in the interaction to achieve shared advantages [33,34];(7) the social affiliation system aims at achieving security and feelings of belongingness in social relationships. It is expressed by thoughts and behaviors related to the experience of feeling part of a group perceived as a source of protection and emotional support [13];(8) the social play system aims at achieving equal interaction through explicit playfulness shared by the individual in group relationships. It could be intended as a precursor of a cooperation system [33].

### 1.2. How to Detect Interpersonal Motivational Systems?

The operationalization of IMSs, as proposed by Liotti [13,16], took a significant step forward with the development of the Assessing Interpersonal Motivations in Transcripts (AIMIT). This coding system was designed to detect the activity of IMSs during therapeutic dialogue [13,19,20] and was based on the systematic transcription of individual adult psychotherapies. AIMIT identified verbal utterances in the participants’ speech that could be indicative of the activation of a given IMS, and has demonstrated good content validity and reliability [19,20]. In clinical practice, AIMIT can be used as a guide for clinicians in evaluating changes during the therapeutic process and serves as a marker of the therapeutic alliance [28,35,36].

Starting from the criteria listed in the AIMIT handbook [13] and further updates [16,18,20], Prevete et al. [21] developed a questionnaire converting AIMIT criteria into one or more statements describing related behaviors and mental states. For instance, a criterion for the submission IMS “inferiority feeling” was converted into the item “I happen to feel inferior to others”. After a subsequential reduction in the number of items, the authors performed an exploratory factor analysis showing a good internal consistency for all IMSs except for the dimension of attachment, which aggregated with the submission dimension, and cooperation, whose items loaded onto different dimensions.

One further proposal for a measure of IMSs was pursued by Brasini et al. [37], who developed a questionnaire called the Social Mentalities Scale (SMS). The questionnaire’s items were organized into seven dimensions (dominance and submission were combined into a single dimension), describing a pattern of affect and action according to a specific social motive. Performing factor analyses, the authors found a six-factor solution: insecurity, prosociality, agonism, belongingness, sexuality, and playfulness, and good indexes of convergent and divergent validity with associated measures [37]. Similarly to Prevete et al. [21], the SMS could not disentangle attachment and submission systems, which collapsed into the common dimension of insecurity, and caregiving and cooperation systems, which collapsed into the common dimension of prosociality.

### 1.3. The Current Study

The current study aimed to develop and validate a new instrument for assessing IMSs in adults, building upon previous attempts to create similar instruments described above. In the first phase, the scale was developed starting from revising the previous measure of IMSs developed by Prevete et al. [21], primarily by adding new items concerning attachment and cooperation, and then exploring the factor structure of the new measure. In the second phase, a confirmatory approach was employed to test the factor structure of the questionnaire and evaluate its construct validity in relation to convergent measures.

## 2. Study I: Development of the Interpersonal Motivational Systems Questionnaire (IMS-Q)

Study I consisted of two steps aimed at constructing the hypothesized eight-factor questionnaire. First, we generated an item pool for each system and then explored the factor structure using Exploratory Factor Analysis (EFA) and Parallel Analysis.

### 2.1. Materials and Methods

#### 2.1.1. Participants

A convenience sample of 455 Italian adults (76.5% females) participated in the study. The sample age ranged from 18 to 80 years (*M* = 34.38, *SD* = 11.37). Most participants had a bachelor’s degree or a higher degree (69%); 26.8% completed the secondary level of education, whereas very few only completed middle school (4.2%).

#### 2.1.2. Procedure

Data were collected in 2018 using an anonymous web-based survey. Participants were contacted through informal channels and invited to disseminate the survey among their contacts, thus activating a snowball sampling procedure. The completion of the survey required approximately 20 min. The study was conducted in accordance with the principles of the Declaration of Helsinki on Ethical Principles for Medical Research and the ethical principles of the Italian Association of Psychology (AIP). Informed consent was obtained from all participants before the administration of the survey, in accordance with the European General Data Protection Regulation (GDPR 2016/679). Inclusion criteria were to be of legal age, not have received a psychiatric diagnosis, and not be in psychotherapeutic treatment.

#### 2.1.3. Measures

Socio-demographic Data. Participants were asked about their sex assigned at birth, age, education level, and geographical provenience.

Initial Item Pool for The Interpersonal Motivational Systems Questionnaire. The questionnaire for assessing the Interpersonal Motivational Systems (Italian translation: Questionario per la Misura dei Sistemi Motivazionali Interpersonali) was developed as follows. We started from a scale previously developed by Prevete et al. [21] consisting of 109 items capturing 6 out of 8 IMSs theorized by Liotti and Monticelli [13]: Affiliation (20 items); Caregiving (15 items); Rank-Dominance (21 items); Sexuality (15 items); Rank-Submission (21 items); and Social Play (17 items). Items for this questionnaire were generated from AIMIT (see the Introduction for a detailed description). Participants in the validation study were asked how frequently they usually experience, during social interactions, such behaviors or mental states using a 4-point Likert scale ranging from “never” (0) to “very often” (3). 

A new set of items was generated for the attachment and cooperation systems since previous attempts failed to identify them as independent factors. To assess attachment, we decided to specifically focus on attachment relationships in adult partnerships. Although attachment style in adults is typically investigated through narration or self-report tools that explore relationships with peers, parents, and partners [38], we chose to focus on romantic love as it is widely acknowledged to be highly influenced by early experiences with caregivers and can therefore be considered a proxy of the construct of interest [39]. Thus, based on the theory that romantic relationships are influenced by early experiences with caregivers, we examined and adapted items from existing validated measures of adult attachment styles and experiences in close relationships (e.g., [40,41]).

For the cooperation system, we followed the theoretical framework proposed by Liotti and Monticelli [35], which draws inspiration from the psycho-evolutionist theory by Tomasello [32]. To select relevant indicators, we focused on those in the AIMIT that relate to the ability to cooperate and emphasize equal treatment. As Tomasello’s theory [32] suggests, true cooperation involves not just working towards a common goal, but also recognizing others’ subjective mental states and intentions, made possible through empathic capacity. This recognition allows us to see others as fellow human beings and facilitates the ability to coordinate reciprocal intentions. Based on these premises, the collaboration system, therefore, emphasizes this ability to coordinate intentions, rather than focusing on social roles and contexts. An example item is “I feel like I’m equal in dignity to others, even though I am helping them”. This feeling is independent of the hierarchy of social roles and is activated by the sense of belonging to the same species due to empathic concern for the conspecifics.

Then, in order to develop a concise and straightforward instrument, we selected only eight items from the original Prevete et al. questionnaire [21], and added eight items for the attachment and cooperation systems, respectively. Items were selected following unidimensionality, clarity, content uniqueness, neutrality, and specificity criteria. Also, since the instrument was intended to be used in generic settings and not just for clinical purposes, we paid particular attention to leaving out those items that operationalized specific traits of psychopathology and personality disorders. The adopted response scale used a 4-point Likert-type approach (0 = ‘never’, 1 = ‘seldom’, 2 = ‘sometimes’, 3 = ‘always’), with no reversed score items.

### 2.2. Analytic Plan

All the analyses were performed in SPSS v28 [42] and R using the lavaan package [43]. The analysis of the 64 items of the IMS-Q began with testing the eight-factor structure described above, using EFA with maximum-likelihood estimation and oblique rotation. Horn’s Parallel Analysis Method [44] was used to establish the correct number of factors. This technique compares the eigenvalues of the observed data to those of randomly simulated data sets to identify the significant factors for interpretation. Factors whose eigenvalue exceeds, at some predetermined probability (95% in this study), the corresponding eigenvalue in the simulative data are considered significant. The goodness of fit was evaluated by assessing the Comparative Fit Index (CFI), Tucker–Lewis index (TLI), Root Mean Square Error of Approximation (RMSEA) and its 90% confidence interval (90% C.I.), standardized Root Mean Squared Residual (SRMR), and Bayesian Information Criterion (BIC). Generally, an acceptable model fit is indicated by values ≥ 0.90 for CFI and TLI, and ≤0.06 for RMSEA and SRMR [45]. The final structure was derived using the following rules: (1) items were considered to load into a factor when the loading was ≥0.30, and (2) items loading into no factors or loading into multiple factors at ≥0.30 were excluded [46]. The BIC was used to compare models, with lower BIC values being preferred.

### 2.3. Results

No items approached skewness > |3| or kurtosis > |10|, indicating that all items were normally distributed [47]. The Parallel Analysis conducted on the initial pool of 64 items supported the hypothesized eight-factor structure (Figure 1A). The model fit of the EFA was generally adequate, CFI = 0.914, RMSEA = 0.035, 90% C.I. [0.032, 0.038], SRMR = 0.030, BIC = 61,106.845. The only exception was the Tucker–Lewis Index (TLI), which stood at 0.89. Additionally, when examining the parameter estimates, six items showed factor loading values less than 0.30. Therefore, they were removed, and an additional EFA and Parallel Analysis were run on the remaining 58 items. The Parallel Analysis confirmed the eight-factor structure (Figure 1B), and the EFA yielded a better model fit, CFI = 0.927, TLI = 0.902, RMSEA = 0.035, 90% C.I. [0.032, 0.038], SRMR = 0.028, BIC = 55,195.553, compared to the previous one. The eight-factor structure was well defined, with all items showing significant and substantial loadings exclusively on their hypothesized factor (Table 1). 

## 3. Study II: Confirmatory Factor Analysis and Construct Validity of the Interpersonal Motivational Systems Questionnaire (IMS-Q)

Study II had two aims. The first aim was to test the factor structure of the questionnaire developed in Study I using the Unrestricted Confirmatory Factor Analysis framework (U-CFA; [48]), and to compare the results with those deriving from the traditional Confirmatory Factor Analysis (CFA) approach. The second aim was to test the convergent validity of the scale by examining the correlations between the IMSs and personality traits [49,50], basic human values [51], and attachment subdimensions [52]. Regarding this second aim, we hypothesized the following significant patterns: (i) positive associations between the caregiving system and the agreeableness personality trait and self-transcendence values; (ii) positive links between the social affiliation system and extraversion and agreeableness personality traits, and self-transcendence values; (iii) positive associations between the attachment system and two dimensions of secure attachment, namely confidence (positive) and discomfort with closeness (negative); (iv) positive associations between the rank-dominance system and self-enhancement values; (v) positive associations between the social play system and extraversion and openness to experience personality traits, and openness to change values; (vi) positive associations between the cooperation system and the agreeableness trait and self-transcendence values, whereas negative associations were expected with the attachment dimension of relationships seen as secondary to achievement; (vii) a positive association between the sexuality system and extraversion; and (viii) a positive association between the rank-submission system and the tendency to seek approval from others. Other potential links were explored.

### 3.1. Materials and Methods

#### 3.1.1. Participants

Data were collected in 2019 using an anonymous web-based survey. A sample of 635 adults (54.8% females) from several Italian regions was recruited through a convenience sampling technique, using the same procedure and criteria as applied in Study I. The sample age ranged from 18 to 80 years (*M* = 44.77, *SD* = 15.71). A total of 44% of the participants had a bachelor’s degree or a higher degree; 44.1% completed the secondary level of education, whereas 11.9% only completed middle school.

#### 3.1.2. Measures

Socio-demographic Data. Participants were asked about their sex assigned at birth, age, education level, and geographical provenience. 

##### Convergent Measures

Big-Five Questionnaire. Personality traits were assessed using the Big Five Questionnaire (BFQ; [50]). The BFQ consists of 60 items, each one describing a personality trait according to the Big Five Factor Model: Extraversion, Agreeableness or Friendliness, Conscientiousness, Emotional stability or Neuroticism, and Intellect or Openness to experience. Participants were asked to agree or disagree with items on the questionnaire on a 5-point scale, ranging from 1 = “strongly disagree” to 5 = “strongly agree”. Cronbach’s alphas (α) supported adequate internal consistency for all scales (α Extraversion = 0.60; α Agreeableness = 0.71; α Conscientiousness = 0.69; α Emotional Stability = 0.86; α Openness to Experience = 0.78).

Portrait Values Questionnaire. The 21-item form of the Portrait Values Questionnaire (PVQ–21; [51,53]) was used. The PVQ–21 measures four higher-order value dimensions: Self-transcendence (universalism and benevolence; Cronbach’s α [5 items] = 0.72); conservation (tradition, conformity, and security; Cronbach’s α [7 items] = 0.70); self-enhancement (power and achievement; Cronbach’s α [4 items] = 0.73) and openness to change (self-direction, stimulation, and hedonism; Cronbach’s α [5 items] = 0.70). Participants were asked to answer the question: “How much like you is this person?” on a 6-point Likert-type scale, ranging from 1 = “not like me at all” to 6 = “very much like me”.

Attachment Style Questionnaire. Adult attachment styles were measured through the 40-item Attachment Style Questionnaire (ASQ, [52]; Italian validation by Fossati et al., [54]). The ASQ is structured along five subscales: confidence in oneself and others (8 items), which refers to the security of attachment; discomfort with closeness (10 items), which refers to the avoidant attachment [55]; relationships as secondary to achievement (7 items), which pertains to the dismissing dimension of attachment style [56]; need for approval (7 items), which reflects the fearful and preoccupied attachment styles [56]; preoccupation with relationships (8 items), which refers to the anxious/ambivalent dimension of attachment style [55]. Items are scored on a 6-point Likert scale ranging from 1 (‘totally disagree’) to 6 (‘totally agree’). Cronbach’s alphas in this study indicated adequate internal consistency for all scales (0.65 for confidence, 0.65 for discomfort with closeness, 0.72 for relationships as secondary to achievement, 0.73 for need for approval, and 0.65 for preoccupation with relationships).

### 3.2. Analytic Plan

After establishing the proper number of factors, U-CFA was applied to the 58 retained items. U-CFA is a confirmatory factor approach that focuses on the pattern of loadings of items on factors. Typically, items in CFA models only load onto their a priori latent factors, with cross-loadings restricted to zero. However, research shows that most items in multidimensional psychological instruments are related to more than one conceptually related factor. Imposing zero cross-loadings often results in poor goodness-of-fit indices, leading researchers to improve them by correlating residual error terms on items, parceling items, or constraining paths to be equal [57]. Unfortunately, these methods overestimate the psychometric validity of instruments, undermining their practical and diagnostic usefulness. In general, some degree of construct-relevant association between items, albeit small, should be expected and small cross-loadings need to be considered to avoid inflated parameter estimates and biased results [57].

To specify U-CFA models, we first identified one anchor item for each hypothesized factor based on a preliminary EFA, selecting the highest loading indicators. The loadings for these items were set to zero for all other factors, while the other indicators were free to load on any factor. Factor variances were fixed to unity, and factor covariances were estimated. To evaluate measurement goodness, we used the cut-off of 0.30 for the inspection of standardized factor loadings (i.e., items that had factor loadings less than 0.30 on their target factor or greater than 0.30 on non-target factors were excluded), and the range of 0.10–0.90 for acceptable residual error variances (i.e., items with residual variances less than 0.10 or greater than 0.90 were removed). Potential differences in model fit were evaluated using BIC, with lower values indicating better fit.

In the second step, in order to ensure model parsimony and reduce complexity, we used the Wald test to identify cross-loadings that could be constrained to zero without significant loss in model fit, and then compared partially unrestricted and fully constrained CFA models for best fit, factor correlations, and parameter estimates. Since all models in this second step are nested in their unrestricted counterparts, differences in model fit were assessed using commonly used benchmarks of ΔRMSEA ≤ 0.015 and ΔCFI ≤ 0.010.

Finally, to assess the construct validity of the IMS-Q, we computed mean scores for each of the eight and then estimated correlations between these scores and convergent measures of personality traits, basic human values, and attachment dimensions.

### 3.3. Results

#### 3.3.1. Confirmatory Factor Analysis

The U-CFA performed on the 58 items resulting from the EFA in Study I presented an adequate model fit, χ^2^ = 2132.230, df = 1217; CFI = 0.920; RMSEA = 0.034, 90% C.I. [0.032, 0.037]; SRMR = 0.027; BIC = 81,594.527, except for TLI, which slightly fell below the established threshold (TLI = 0.89). Also, when looking at the parameter estimates, the solution showed that four items (see items 6, 20, 21, and 22 in Table 1) had loadings lower than 0.30 on their target factor. Three of these items also presented significant loadings higher than 0.30 on non-target factors (6, 21, and 22). Furthermore, four items presented cross-loadings exceeding 0.30 (see items 12, 23, 33, and 56 in Table 1). As a result of the aforementioned criteria, we excluded these items and ran the analysis again. The model without these items showed an overall improvement in model fit, as indicated by the lower BIC value (BIC = 70,867.820; overall model fit: χ^2^ = 1499.060; df = 853; CFI = 0.932; TLI = 0.905; RMSEA = 0.035, 90% C.I. [0.032, 0.037]; SRMR = 0.025). All items presented factor loadings higher than 0.30, and no critical cross-loadings were detected. Residual variances ranged from 0.371 (item 24) to 0.746 (item 2).

As a second step, we moved to inspect nonsignificant cross-loadings that could be constrained to zero. Two hundred ninety-six loadings were nonsignificant and then constrained to zero. These changes did not result in a decline in fit, χ^2^ = 1842.7565, df = 1082; CFI = 0.922; TLI = 0.912; RMSEA = 0.033, 90% C.I. [0.031, 0.036], SRMR = 0.036, BIC = 69,680.579, ΔRMSEA = 0.002, and ΔCFI = 0.010. The standardized factor loadings and reliability indicators of the model are reported in Table 2. Additionally, both Cronbach’s alphas and McDonald’s omegas indicated satisfactory internal consistency [58].

Finally, we ran a traditional CFA and compared it to the factor structure resulting from the final U-CFA using the criteria introduced earlier. Overall, the CFA model does not achieve a reasonable fit by conventional standards, χ^2^ = 2389.191, df = 1147; CFI = 0.873; TLI = 0.864; RMSEA = 0.041, 90% C.I. [0.039, 0.044], SRMR = 0.054, BIC = 69,869.585, and the change in CFI exceeds the frequently used benchmarks of ΔCFI ≤ 0.010. These findings indicated that the U-CFA model, despite having more free parameters, demonstrated a more satisfactory fit to the data than the traditional CFA model. This supports the notion that permitting the identified cross-loadings within the U-CFA model better captured the underlying structure of the data.

#### 3.3.2. Convergent Validity

When analyzing the associations between the IMSs and convergent measures, the results generally supported our hypotheses, with also some notable non-hypothesized associations (Table 3). Specifically, (i) the caregiving system was positively associated with agreeableness and self-transcendence values; (ii) social affiliation was positively associated with extraversion and agreeableness traits, and self-transcendence values. Notably, we also found a positive association with openness to experience trait and secure attachment as represented by the confidence dimension, and a negative association with discomfort with closeness; (iii) attachment was positively linked to confidence, and negatively associated with discomfort with closeness; (iv) rank-dominance was positively associated with self-enhancement values. Interestingly, we found a significant association with openness to experience trait and relationships as secondary to achievement; (v) social play was positively associated with extraversion, openness to experience trait, and openness to change values; (vi) cooperation was positively associated with agreeableness trait and self-transcendence values, but negatively related to relationships as secondary to achievement; and (vii) sexuality was positively linked to extraversion, as hypothesized. Interestingly, we also found positive relationships with self-enhancement and openness to change values and a negative link with conservation. Finally, the rank-submission system showed a positive relationship with the need for approval. Notably, negative associations with emotional stability, openness to experience, conscientiousness, and confidence were also found, along with a positive association with the preoccupation with relationships.

## 4. Discussion

This study aimed to develop a new questionnaire that measures the eight IMSs as theorized by Liotti and collaborators [13,16]: attachment, caregiving, rank-dominance, rank-submission, sexuality, cooperation, social affiliation, and social play. Liotti’s model stems from previous theoretical contributions [10,11] and found a specific application in clinical practice. Additionally, the study contributes to enhancing our understanding on the connections among motivational systems, values, and personality. 

Motivational systems are believed to have emerged during the evolutionary process, enabling individuals to interact with one another in an adaptive way. They are biologically based and selectively activated by specific external or internal cues. However, despite the growing interest in their adaptive function [4], empirical measures of motivational systems are lacking. In our study, we developed a self-report instrument able to capture the eight IMSs. Content validity was proved by explorative and confirmatory factor analyses evidencing that each IMS was independent of the others. Empirical validity was established by finding correlations between the IMSs and other measures that were consistent with our theoretical assumptions and expected results. 

The main steps of the study included reducing the number of items from the previous measure by Prevete et al. [21], generating a new pool of items related to the attachment and cooperation systems not identified in the previous version, performing factor analyses to ensure adequacy, and testing convergent validity. Attachment items were selected focusing on the context of romantic relationships, which is a reliable dimension to measure attachment in adulthood, rather than considering experiences of danger, loneliness, and illness coded in AIMIT as an expression of the attachment system. To identify items related to the cooperation system, we drew on Tomasello’s theory [32], which posits that collaboration is a specific attitude of the human species concerning the intentional disposition of individuals to pursue common purposes with other individuals. 

The factor analyses led to the definition of a questionnaire including 50 items that adequately identified the eight theorized IMSs. The questionnaire met the criteria of simplicity, clarity, and uniqueness of each dimension, and is a promising instrument for assessing how frequently and intensively individuals activate each IMS. 

The correlations with convergent measures were generally consistent with the content validity of each IMS. Specifically, we found that the caregiving system was most strongly associated with the personality traits of agreeableness and self-transcendence values, which supports the thesis that individuals who are highly concerned for the well-being of others perceived as vulnerable [29] are more likely to activate their caregiving system, as already found by Brasini et al. [37]. Similarly, the social affiliation system also showed associations with agreeableness and self-transcendence but was additionally associated with the personality trait of openness to experience and the attachment dimension of confidence. These findings suggest that the social affiliation system is characterized by a willingness to engage in social exchanges within secure and close relationships.

Identifying the attachment system in isolation from others remains challenging due to the fact that both strategies of hyperactivation (eliciting anxiety/preoccupation responses) and deactivation (triggering avoidance responses) disclose the functioning of the attachment system [59]. Furthermore, as previously mentioned, earlier studies encountered difficulty in differentiating attachment from closely related systems. We acknowledge that our decision to exclusively focus our investigation on the realm of romantic love may pose a limitation, but it enabled us to set it apart from other dimensions. However, our data displayed notable correlations with attachment subdimensions such as confidence (in a positive direction) and discomfort with closeness (in a negative direction).

It is noteworthy that the convergent associations of the two ranking dimensions, dominance and submission, were not symmetric, confirming the independence of these two systems. The only clearly symmetric association was with openness to experience (positive for dominance and negative for submission). The other associations were domain-specific. The dominance system was positively related to the value of self-enhancement, which represents the pursuit of one’s interests and relative success and dominance over others, as emphasized by Schwartz [51]. Furthermore, the dominance system was associated with the view of relationships as secondary to achievement. Contrarily, the submission system showed the highest association with lack of confidence, need for approval, and emotional instability. In other terms, insecurity seems to be a personal characteristic associated with the submission system, whereas the importance given to power and openness to experience seems to be the salient characteristics of the dominance system.

The social play system, as defined by Panksepp [6], refers to the innate tendency of individuals to engage in playful physical contact with others in a joyful and playful context. Our study found that this system had similar associations as the social affiliation system, such as extraversion and openness to experience, but without significant implications in terms of prosocial behavior tendencies (e.g., agreeableness trait and self-transcendence values). These findings support the idea that the unique function of the social play system is to facilitate playfulness among individuals in group relationships.

The peculiar association of the cooperation system is with self-transcendence values, confirming the moral nature of this system as theorized by Tomasello [32], such as the prominence of the search for good relationships with others rather than the pursuit of personal success. The importance of an innate disposition to cooperate with each other is introduces a new and interesting perspective to the study of motivation. It is not only present in toddlers who help an adult in distress but also in other mammalians [60] above and beyond the risk of social desirability in an individual’s responses. Lastly, the sexual system was found to be highly correlated with self-enhancement, similar to the dominance system. However, we also observed a strong association with the drive to search for new and exciting experiences.

Although the present study aimed to test the psychometric properties of a new measurement tool for IMSs, the findings also offer an opportunity for some theoretical considerations. One consideration pertains to the number of dimensions related to motivational systems. We recognize that various models propose distinct classifications. This is the case, for instance, of Panksepp’s model, which hypothesizes a link between motivations and primary emotional systems shared across all mammalian species [61], or Dweck’s model, which identifies the “basic” human motives of predictability, competence, and acceptance [5]. Furthermore, the relationship between motivational systems and other structural facets of personality, like the Big Five, remains an open question. In this context, Del Giudice’s [14] suggestions are noteworthy, as they differentiate between the structural approach to personality (based on factorial analysis) and the functional approach to personality traits that closely interact with motivational systems. In this perspective, individual differences could be examined through the lens of how flexible or rigid the various motivational systems are.

The current study has several key strengths, including satisfactory indexes of fit-in factor analyses that effectively supported the independence of the eight IMSs, content validity supported by convergent analyses, and a reduced number of items that facilitate the ease and practicality of using the questionnaire (see Appendix A for the Italian version of the instrument). However, the study is not without limitations. The use of convenience sampling is a potential source of sampling bias, which could negatively impact the internal validity of the study. Furthermore, we did not consider the potential impact of social desirability, which could have influenced how individuals responded to self-reports. We also did not account for other socio-demographic information about the participants (e.g., marital status), which could have introduced biases into the results. Further, our study’s focus on attachment was confined solely to romantic love, which inherently limits its ability to encompass the diverse array of situations that arise from the multifaceted nature of attachment motivation. Also, it should be noted that IMSs predominantly operate outside of consciousness, whereas individuals are required to evaluate the relevance of systems based on conscious evaluations. According to Del Giudice [14], translating IMSs into unique outputs can be difficult, as a single motivation can lead to different behaviors. For instance, attachment needs can result in an intense search for closeness but also withdrawal and detachment.

Despite these limitations, the investigation of IMS through the IMS-Q is a promising area of research that will shed light on a new area of research in personality studies and clinical practice. Future studies should further investigate the psychometric properties of the measure and test for socio-cultural and cross-cultural invariance, including age and gender. One area that requires further exploration concerns the stability and change in IMS over time. Although the theory posits relative stability of the IMS within individuals, it is also plausible that significant life events, such as psychotherapeutic processes, could alter the priority assigned to the different motivational systems and activate systems that are typically under-expressed. Conversely, it can be assumed that individuals may be fixed on a single system over time, rather than activating them all flexibly. Only longitudinal studies can provide insights into the temporal trends of IMS. Finally, since the theory of the IMS stems from the clinical observations of patients in psychotherapy, the most promising area for exploring the IMS questionnaire is its usability in clinical research. Since psychopathology may be related to unbalanced relations among motivational systems, namely the predominance of one or more motivational systems over the others, it could be crucial to investigate flexibility versus rigidity as a clinical index to evaluate mental health in individuals. Overall, these findings suggest that the IMS framework can be used to understand individual differences in motivation and behavior in different social contexts.

## Figures and Tables

**Figure 1 behavsci-13-00784-f001:**
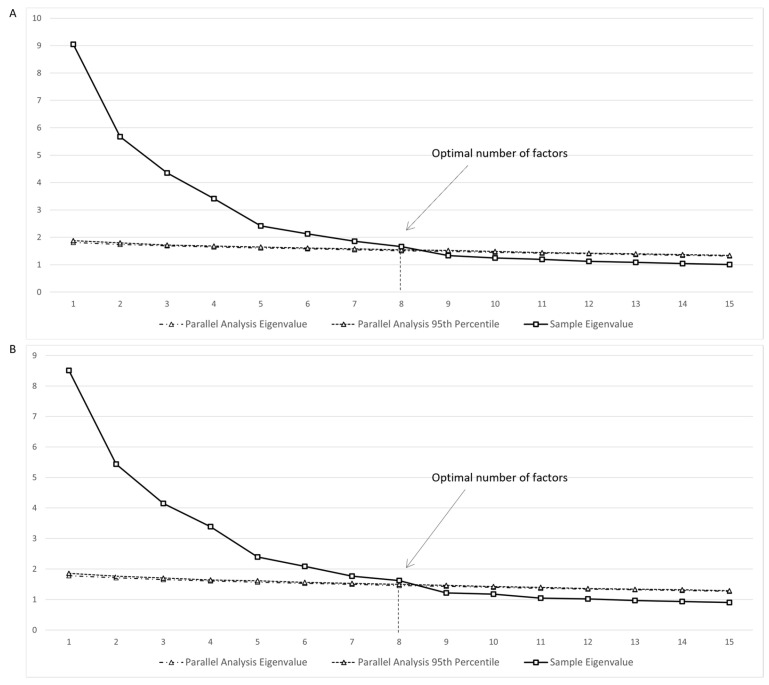
Results from the Parallel Analysis. (**A**) Initial EFA Model and Parallel Analysis (64 Items). (**B**) Refined EFA Model and Parallel Analysis (58 Items). For the sake of clarity and simplicity, only eigenvalues corresponding to up to 15 factors were presented.

**Table 1 behavsci-13-00784-t001:** Results from the Exploratory Factor Analysis (N = 455).

IMS	Item Content (Item Number)	Mean	Skewness	Kurtosis	Factor 1	Factor 2	Factor 3	Factor 4	Factor 5	Factor 6	Factor 7	Factor 8
Caregiving	I happen to help others because I feel they need it (1)	2.31	−0.45	−0.13	**0.72**	0.02	−0.04	0.06	0.00	0.01	0.03	0.04
Caregiving	I happen to feel the urge to take care of others and their needs (2)	2.28	−0.66	0.35	**0.76**	−0.07	0.13	0.01	−0.10	0.04	0.11	−0.01
Caregiving	I happen to take care of others, though they don’t ask for it (3)	2.10	−0.32	−0.47	**0.75**	−0.02	−0.06	−0.01	0.04	0.01	−0.06	−0.01
Caregiving	I happen to provide others with comfort, assurance, and protection (4)	2.45	−0.82	0.55	**0.46**	0.27	−0.02	0.05	−0.03	0.02	0.03	−0.11
Caregiving	I find myself worrying about others and the risks they can take (5)	2.14	−0.50	−0.08	**0.48**	0.09	−0.04	0.17	−0.01	0.03	−0.01	0.06
Caregiving	I feel like it’s my fault if I disregard or don’t meet others’ needs (6)	2	−0.40	0.07	**0.51**	0.07	0.07	−0.05	0.08	−0.14	−0.02	0.25
Caregiving	I worry and get active if someone is in danger (7)	2.52	−0.60	−0.72	**0.45**	0.20	−0.01	−0.05	0.04	0.13	0.01	−0.04
Social Affiliation	When I’m with my colleagues/collaborators/friends, I feel as if we share the same goals (8)	1.91	−0.40	0.83	−0.01	**0.65**	0.08	0.05	−0.02	0.07	−0.03	0.01
Social Affiliation	When I compare my interests with those of the members of my group (e.g., colleagues, friends, collaborators, and other groups), I can see some affinities (9)	1.92	−0.17	0.20	−0.13	**0.58**	0.05	−0.01	−0.06	0.23	0.00	0.08
Social Affiliation	When I’m with my friends, I happen to feel like part of the group (10)	2.24	−0.56	−0.15	0.03	**0.56**	0.02	−0.06	0.13	0.03	0.00	−0.17
Social Affiliation	When I partake in activities with the members of a group, I happen to feel the pleasure of doing it (11)	2.28	−0.30	0.12	0.10	**0.32**	0.13	0.05	0.06	0.28	0.05	−0.05
Social Affiliation	I actively engage in maintaining strong bonds among the members of my group (family, friends, class, and other groups) (12)	2.15	−0.47	−0.37	0.21	**0.47**	−0.02	−0.04	0.09	0.07	−0.01	0.06
Social Affiliation	I happen to use the terms “us” or “we” to refer to my group (e.g., class, team, fellow citizens, clubs, friends, and other groups) (13)	2.33	−0.88	−0.01	0.14	**0.43**	0.11	0.12	0.04	−0.02	0.02	−0.06
Social Affiliation	I happen to feel like I’m part of a wider group (e.g., “we women,” “we men,” “we young people,” “we elders,” etc.) (14)	1.61	−0.07	−0.46	0.08	**0.31**	0.16	0.06	0.04	0.05	0.10	0.00
Attachment	I happen to feel my partner close to me when I need it (15)	2.27	−0.86	0.02	−0.01	0.00	**0.83**	−0.01	−0.01	0.05	0.04	0.05
Attachment	I happen to believe my partner shows me the love I deserve (16)	2.11	−0.73	−0.15	−0.06	−0.02	**0.82**	−0.06	0.05	−0.06	0.00	−0.03
Attachment	I happen to feel my partner supports me when I need it (17)	2.11	−0.68	−0.35	0.00	0.02	**0.70**	0.05	−0.03	0.08	−0.04	0.17
Attachment	I happen to trust my partner (18)	2.42	−1.30	1.12	0.02	0.01	**0.80**	−0.04	0.04	0.01	−0.06	0.03
Attachment	I happen to feel like my partner won’t leave me (19)	1.85	−0.49	−0.85	−0.06	0.01	**0.70**	0.00	0.03	−0.05	−0.10	−0.05
Attachment	I feel comfortable when being cuddled (20)	2.22	−0.75	−0.05	0.11	0.11	**0.40**	−0.03	−0.07	0.03	0.18	−0.06
Attachment	I happen to feel satisfied with the love I get from others (21)	2.03	−0.465	0.33	0.02	0.18	**0.41**	0.03	−0.03	0.23	0.04	−0.10
Dominance	I happen to criticize others (22)	1.52	0.20	−0.23	−0.05	−0.18	0.02	**0.48**	0.00	0.10	0.03	0.24
Dominance	When I’m mad at some other people, I happen to scream and rant (23)	1.42	0.16	−0.62	0.04	0.04	0.03	**0.54**	−0.10	−0.04	0.04	0.05
Dominance	I happen to think that people are less able than me (24)	1.15	0.51	0.44	0.03	−0.26	0.01	**0.43**	0.14	0.05	0.07	−0.08
Dominance	I happen to give permissions and prohibitions to others (25)	0.98	0.63	−0.10	0.08	0.19	−0.02	**0.50**	0.09	−0.09	−0.08	0.06
Dominance	I like to feel self-satisfied with my successes and show others my victories (26)	1.04	0.42	−0.38	0.00	−0.04	−0.05	**0.38**	0.07	0.01	0.22	−0.03
Dominance	I happen to take control of situations (27)	1.76	−0.01	−0.65	0.23	0.08	0.00	**0.33**	0.13	−0.02	−0.05	−0.29
Dominance	I happen to verbally frighten others (28)	0.83	0.69	0.29	−0.03	0.02	−0.18	**0.57**	−0.01	0.03	0.22	−0.01
Social play	I like to tell fun stories/jokes just for fun purposes (29)	1.46	0.03	−0.86	−0.03	0.01	−0.06	−0.01	**0.70**	−0.05	0.00	−0.12
Social play	I happen to tell playful stories to play it down, but without ridiculing the other person (30)	1.75	−0.23	−0.68	0.04	−0.03	0.04	0.01	**0.66**	0.00	0.11	−0.04
Social play	I find myself making jokes that others enjoy, without offending anyone (31)	2.01	−0.53	−0.05	0.04	−0.02	0.07	0.05	**0.73**	0.12	−0.04	0.01
Social play	I happen to be playful and friendly with others (32)	2.27	−0.60	0.22	0.03	0.16	0.02	−0.02	**0.66**	0.06	0.08	0.02
Social play	I happen to highlight the comic sides of the events, without any intention of offending anyone (33)	1.99	−0.33	−0.47	−0.05	−0.02	−0.09	0.06	**0.60**	0.18	0.00	0.05
Social play	I happen to alter my voice to have fun with others (e.g., I change my voice, I do strange or funny voices, etc.) (34)	1.75	−0.20	−1.01	−0.02	−0.02	0.13	0.00	**0.51**	−0.09	0.02	0.03
Social play	I happen to organize funny jokes, but not hurtful (35)	1.08	0.46	−0.46	−0.06	0.11	0.01	0.05	**0.43**	−0.06	0.15	0.01
Social play	I can propose funny games/activities to others, without imposing myself (36)	1.67	−0.26	−0.40	0.06	0.24	−0.06	−0.05	**0.39**	0.00	0.02	−0.05
Cooperation	I happen to feel equal to others, on the same plane of dignity (37)	1.91	−0.35	−0.15	−0.09	0.05	−0.03	0.02	0.02	**0.80**	−0.03	−0.13
Cooperation	I like to feel having equal dignity while interacting with another person (38)	2.46	−0.68	0.24	0.14	−0.09	0.07	0.01	0.02	**0.59**	0.10	0.07
Cooperation	I feel like I’m equal in dignity to others, even though I’m helping them (39)	2.55	−0.87	0.17	0.14	0.01	0.01	−0.23	0.02	**0.49**	0.01	0.08
Cooperation	I feel like I’m equal in dignity to others, even though I’m asking them for help (40)	2.24	−0.56	−0.10	−0.01	0.02	0.10	0.02	−0.02	**0.57**	−0.07	−0.21
Cooperation	I happen to think that I’m on the same plane as another person, independently of our social roles (41)	2.40	−0.95	0.76	0.10	0.11	−0.01	−0.21	0.08	**0.35**	−0.01	−0.03
Cooperation	I happen to think I have interchangeable roles with others, since, as humans, we have the same dignity (42)	2.07	−0.44	0.15	0.18	0.18	−0.04	−0.06	0.00	**0.41**	0.00	−0.08
Sexuality	I find myself imaging sexual scenes with real and/or imaginary partners that excite me (43)	1.40	0.09	−1.06	0.09	−0.19	0.03	−0.08	0.09	0.02	**0.68**	0.11
Sexuality	I happen to make seductive compliments (44)	0.85	0.70	−0.07	0.01	0.04	−0.05	0.06	0.00	−0.09	**0.77**	−0.09
Sexuality	I happen to be sexually attracted to others (45)	1.36	0.24	−0.54	0.04	−0.12	0.01	−0.06	0.04	0.05	**0.70**	0.13
Sexuality	I happen to go out to seduce or sexually provoke people (46)	0.75	0.92	0.35	−0.09	0.02	−0.06	0.09	−0.02	−0.01	**0.72**	0.04
Sexuality	I happen to tell others about my sexual attractions, without any shame (47)	1.05	0.50	−0.59	−0.05	0.03	0.01	0.01	0.13	0.07	**0.52**	−0.07
Sexuality	I happen to accept, without embarrassment, erotic compliments (48)	1.09	0.38	−0.56	−0.07	0.06	−0.01	−0.03	0.00	−0.04	**0.65**	−0.20
Sexuality	I happen to be seduced by seductive attitudes or ways of dressing (49)	1.27	0.34	−0.64	0.03	−0.15	−0.01	0.12	−0.01	0.00	**0.61**	0.02
Sexuality	I happen to notice that people look at me seductively (50)	0.96	0.57	0.15	0.03	0.04	0.03	0.15	−0.01	−0.03	**0.43**	−0.20
Submission	I happen to feel inferior to others (51)	1.47	0.23	−0.38	0.08	−0.08	−0.13	0.00	−0.03	0.00	−0.07	**0.64**
Submission	I happen to avoid competition for fear of receiving a negative judgment from others (52)	1.34	0.28	−0.86	−0.06	−0.12	0.05	0.05	0.03	0.01	0.02	**0.66**
Submission	I happen to get really ashamed when I make a fool of myself (53)	2.13	−0.52	−0.48	0.20	0.00	−0.02	0.05	−0.04	−0.04	−0.06	**0.60**
Submission	When someone shouts at me, I happen to look down (54)	1.13	0.44	−0.54	0.03	0.08	0.04	−0.13	−0.05	−0.06	0.07	**0.59**
Submission	I prefer to have someone to guide me and tell me what to do when facing important decisions (55)	1.36	0.19	−0.49	−0.08	0.14	0.02	0.07	0.02	−0.07	−0.04	**0.61**
Submission	I ask for advice from others before doing something because I consider them better than me (56)	1.23	0.49	0.24	−0.01	0.06	−0.02	−0.04	−0.04	0.04	0.00	**0.58**
Submission	When someone verbally takes me back, I feel humiliated/scorned (57)	1.69	−0.24	−0.48	0.11	−0.05	−0.04	0.04	0.04	−0.05	−0.02	**0.72**
Submission	I happen to give up easily in disputes (58)	1.13	0.44	−0.33	−0.10	0.08	−0.02	−0.15	0.00	0.00	0.07	**0.55**

Note. Factor loadings in bold indicate expected significant loadings at *p* < 0.001.

**Table 2 behavsci-13-00784-t002:** Results from the Unrestricted Confirmatory Factor Analysis (N = 635).

IMS	Item Content	Factor 1	Factor 2	Factor 3	Factor 4	Factor 5	Factor 6	Factor 7	Factor 8	Error Variance
Caregiving	I happen to help others because I feel they need it	**0.71 *****	0.00	−0.13 **	0.00	0.00	0.00	0.00	0.00	0.52
Caregiving	I happen to feel the urge to take care of others and their needs	**0.71 *****	0.00	0.00	0.00	0.00	0.00	−0.06	0.00	0.49
Caregiving	I happen to take care of others, though they don’t ask for it	**0.67 *****	0.00	0.00	0.00	0.00	0.00	0.00	0.00	0.56
Caregiving	I happen to provide others with comfort, assurance, and protection	**0.62 *****	0.14 **	0.00	0.00	0.00	0.00	0.00	0.00	0.52
Caregiving	I find myself worrying about others and the risks they can take	**0.45 *****	0.00	0.00	0.00	0.00	0.00	−0.14 **	0.09 *	0.75
Caregiving	I worry and get active if someone is in danger	**0.63 *****	0.00	0.00	0.00	0.00	0.00	0.00	0.00	0.61
Social Affiliation	When I’m with my colleagues/collaborators/friends, I feel as if we share the same goals	0.00	**0.50 *****	0.00	0.00	0.00	0.00	0.00	0.00	0.76
Social Affiliation	When I compare my interests with those of the members of my group (e.g., colleagues, friends, collaborators, and other groups), I can see some affinities	0.00	**0.61 *****	0.00	0.00	0.00	0.00	0.00	0.00	0.63
Social Affiliation	When I’m with my friends, I happen to feel like part of the group	0.00	**0.54 *****	0.00	0.00	0.16 **	0.00	0.00	−0.14 **	0.59
Social Affiliation	When I partake in activities with the members of a group, I happen to feel the pleasure of doing it	0.00	**0.63 *****	0.00	0.00	0.11 *	0.00	0.00	0.00	0.56
Social Affiliation	I happen to use the terms “us” or “we” to refer to my group (e.g., class, team, fellow citizens, clubs, friends, and other groups)	0.00	**0.47 *****	0.00	0.00	0.00	0.16 *	0.00	0.13 **	0.67
Social Affiliation	I happen to feel like I’m part of a wider group (e.g., “we women,” “we men,” “we young people,” “we elders,” etc.)	0.00	**0.46 *****	0.00	0.09 *	0.00	0.00	0.00	0.17 **	0.74
Attachment	I happen to feel my partner close to me when I need it	0.00	0.00	**0.70 *****	0.00	0.00	0.00	0.09 *	0.00	0.51
Attachment	I happen to believe my partner shows me the love I deserve	0.00	0.00	**0.73 *****	0.00	0.00	0.00	0.00	0.00	0.47
Attachment	I happen to feel my partner supports me when I need it	0.00	0.00	**0.81 *****	0.00	0.00	0.00	0.00	0.09 *	0.37
Attachment	I happen to trust my partner	0.00	0.00	**0.74 *****	0.00	0.00	0.00	0.00	0.00	0.45
Attachment	I happen to feel like my partner won’t leave me	0.00	0.00	**0.69 *****	0.00	0.00	0.00	0.00	0.11 **	0.54
Dominance	I happen to think that people are less able than me	−0.10 *	−0.29 ***	0.11 *	**0.47 *****	0.00	0.00	0.20 ***	0.11 *	0.66
Dominance	I happen to give permissions and prohibitions to others	0.00	0.00	0.00	**0.54 *****	−0.12 *	0.00	0.00	0.16 **	0.72
Dominance	I like to feel self-satisfied with my successes and show others my victories	−0.11 *	0.00	0.00	**0.41 *****	0.00	0.00	0.21 ***	0.01 *	0.76
Dominance	I happen to take control of situations	0.00	0.00	0.00	**0.62 *****	0.00	0.00	0.00	0.00	0.61
Dominance	I happen to verbally frighten others	0.00	−0.27 ***	0.00	**0.49 *****	0.00	0.00	0.13 *	0.15 **	0.69
Social Play	I like to tell fun stories/jokes just for fun purposes	0.00	0.00	0.00	0.00	**0.77 *****	0.00	0.00	0.00	0.40
Social Play	I happen to tell playful stories to play it down, but without ridiculing the other person	0.00	0.00	0.00	0.00	**0.67 *****	0.00	0.00	0.12 **	0.55
Social Play	I find myself making jokes that others enjoy, without offending anyone	0.00	0.00	0.00	0.00	**0.78 *****	0.00	0.00	0.00	0.39
Social Play	I happen to be playful and friendly with others	0.18 **	0.00	0.00	0.00	**0.60 *****	0.00	0.00	−0.08 *	0.55
Social Play	I happen to alter my voice to have fun with others (e.g., I change my voice, I do strange or funny voices, etc.)	0.00	0.00	0.00	0.00	**0.40 *****	0.00	0.16 **	0.11 *	0.76
Social Play	I happen to organize funny jokes, but not hurtful	0.00	0.00	0.00	0.00	**0.55 *****	0.00	0.22 ***	0.00	0.57
Social Play	I can propose funny games/activities to others, without imposing myself	0.00	0.14 **	0.00	0.00	**0.46 *****	0.00	0.19 **	0.00	0.64
Cooperation	I happen to feel equal to others, on the same plane of dignity	0.00	0.00	0.00	0.00	0.00	**0.47 *****	0.00	0.00	0.78
Cooperation	I like to feel having equal dignity while interacting with another person	0.00	0.00	0.00	0.00	0.00	**0.44 *****	0.00	0.20 ***	0.78
Cooperation	I feel like I’m equal in dignity to others, even though I’m helping them	0.09 *	0.00	0.00	−0.11 *	0.00	**0.65 *****	0.00	0.00	0.56
Cooperation	I feel like I’m equal in dignity to others, even though I’m asking them for help	0.00	0.00	0.00	0.00	0.00	**0.64 *****	0.00	0.00	0.59
Cooperation	I happen to think that I’m on the same plane as another person, independently of our social roles	0.00	0.00	0.00	−0.11 *	0.00	**0.72 *****	0.00	0.00	0.52
Cooperation	I happen to think I have interchangeable roles with others, since, as humans, we have the same dignity	0.00	0.00	0.00	0.00	0.00	**0.49 *****	0.16 ***	0.08 *	0.72
Sexuality	I find myself imaging sexual scenes with real and/or imaginary partners that excite me	0.00	0.00	0.00	0.00	0.00	0.00	**0.66 *****	0.00	0.56
Sexuality	I happen to make seductive compliments	0.00	0.00	0.00	0.00	0.00	0.00	**0.66 *****	0.00	0.57
Sexuality	I happen to be sexually attracted to others	0.00	0.00	0.00	0.00	0.00	0.00	**0.72 *****	0.00	0.48
Sexuality	I happen to go out to seduce or sexually provoke people	0.00	0.14 **	−0.10 **	0.00	−0.09 *	0.00	**0.72 *****	0.00	0.51
Sexuality	I happen to tell others about my sexual attractions, without any shame	0.00	0.00	0.00	0.00	0.00	0.14 **	**0.64 *****	0.00	0.56
Sexuality	I happen to accept, without embarrassment, erotic compliments	0.00	0.00	0.00	0.00	0.00	0.00	**0.68 *****	0.00	0.53
Sexuality	I happen to be seduced by seductive attitudes or ways of dressing	0.00	0.00	0.00	0.00	0.00	0.00	**0.74 *****	0.00	0.45
Sexuality	I happen to notice that people look at me seductively	0.00	0.00	0.00	0.20 ***	0.00	0.00	**0.49 *****	0.00	0.67
Submission	I happen to feel inferior to others	0.00	0.00	0.00	−0.25 ***	0.00	0.00	0.00	**0.60 *****	0.57
Submission	I happen to avoid competition for fear of receiving a negative judgment from others	0.00	0.00	0.09 *	−0.25 ***	0.00	0.00	0.00	**0.69 *****	0.47
Submission	I happen to get really ashamed when I make a fool of myself	0.00	0.00	0.00	0.00	0.00	0.00	0.00	**0.49 *****	0.76
Submission	When someone shouts at me, I happen to look down	0.00	0.00	0.17 **	−0.14 **	0.00	0.00	0.00	**0.55 *****	0.68
Submission	I prefer to have someone to guide me and tell me what to do when facing important decisions	0.00	0.00	0.12 **	0.00	0.00	0.00	0.00	**0.49 *****	0.76
Submission	I happen to get really ashamed when I make a fool of myself	0.00	0.00	0.00	0.00	0.00	0.00	0.00	**0.64 *****	0.59
Submission	I happen to give up easily in disputes	0.00	0.00	0.00	−0.21 ***	0.00	0.00	0.00	**0.53 *****	0.67
*Reliability coefficients*										
Cronbach’s Alpha		0.80	0.73	0.85	0.65	0.83	0.73	0.87	0.79	
McDonald’s Omega		0.81	0.73	0.85	0.65	0.83	0.73	0.86	0.80	

Note. * *p* < 0.05; ** *p* < 0.01; *** *p* < 0.001. Factor loadings in bold indicate expected significant loadings.

**Table 3 behavsci-13-00784-t003:** Correlations between the eight IMSs and the hypothesized convergent measures.

	IMS							
	Caregiving	Social Affiliation	Attachment	Dominance	Social Play	Cooperation	Sexuality	Submission
*Personality traits*								
Extraversion	0.15 ***	0.24 ***	0.07	0.05	0.24 ***	0.23 ***	0.25 ***	−0.25 ***
Emotional stability	−0.05	0.18 ***	0.09 *	−0.18 ***	0.03	0.10 *	−0.01	−0.30 ***
Agreeableness	0.40 ***	0.44 ***	0.11 **	−0.20 ***	0.18 ***	0.27 ***	−0.07	−0.05
Openness to experience	0.16 ***	0.32 ***	0.09 *	0.34 ***	0.33 ***	0.12 **	0.24 ***	−0.35 ***
Conscientiousness	0.16 ***	0.23 ***	0.08	0.11 *	0.03	0.11 **	−0.03	−0.25 ***
*Basic values*								
Self-transcendence	0.47 ***	0.40 ***	0.10 **	−0.08 *	0.22 ***	0.44 ***	−0.02	0.00
Self-enhancement	−0.09 *	0.04	0.07	0.52 ***	0.13 ***	0.02	0.41 ***	0.03
Conservation	0.26 ***	0.19 ***	0.08 *	−0.03	−0.02	0.05	−0.24 ***	0.07
Openness to change	0.14 ***	0.21 ***	0.03	0.24 ***	0.30 ***	0.20 ***	0.36 ***	−0.12 **
*Attachment*								
Confidence	0.22 ***	0.43 ***	0.32 ***	0.06	0.23 ***	0.24 ***	−0.03	−0.33 ***
Discomfort with closeness	−0.15 ***	−0.28 ***	−0.23 ***	0.09 *	−0.12 **	−0.16 ***	0.07	0.17 ***
Relationships as secondary to achievement	−0.29 ***	−0.24 ***	−0.14 ***	0.24 ***	−0.09 *	−0.29 ***	0.13 ***	0.15 ***
Need for approval	−0.05	−0.20 ***	−0.13 ***	−0.02	−0.06	−0.20 ***	0.09 *	0.64 ***
Preoccupation with relationships	0.12 **	−0.09 *	0.01	0.07	−0.02	−0.07	0.01	0.38 ***

Note. * *p* < 0.05; ** *p* < 0.01; *** *p* < 0.001.

## Data Availability

The data presented in this study are available on request from the corresponding author.

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
