# Peer review of "Development and Validation of the Interpersonal Motivational Systems Questionnaire (IMS-Q)"

_behavsci, 2023, doi:10.3390/bs13090784_

Round 1

Reviewer 1 Report

Dear Authors,

Congratulations on your submission. After reviewing your manuscript, my comments are mainly limited to a few considerations on the methods, and some things I believe missing. Otherwise, the rest of the comments are minor changes and suggestions I believe the authors will be able to handle without any problem.

I have two major concerns: the absence of the TLI estimator in the analyses, and the lack of literature discussed in the discussion section. The former I explain more below; however, it is concerning that the estimator is not reported. The latter is simply unacceptable, and denotes the manuscript requires major changes in this section. The study results should not be offered by themselves. They co-exist within the literature. That is why the section is called that way, to enhance interpretability of the results and their impact.

Below you may find these comments and suggestions, broken down by sections.

Abstract

Line 17 = “fill this gap” sounds like the study does it all. “Contribute to fill the gap” seems more appropriate.

Introduction

The first two paragraphs discus classical, yet very old literature on the topic. It is up the authors to consider removing them, for it they are not wrong. It just does not add anything to the article if taken out. Besides, it is a validation study, going that far in the theory is not justified. Of course, if taken out, the new first paragraph would need an appropriate new first sentence.

Line 57 = I believe the authors refer to “innate” rather than “inmate” rules.

Methods

It is weird that I do not see the Tucker–Lewis index (TLI) index. This is perhaps a must in exploratory factor analysis. TLI evaluates the incremental improvement in fit of a given substantive model over that of a null model. Moreover, this statistic is often given along with the CFI.

Widaman, K. F., & Thompson, J. S. (2003). On specifying the null model for incremental fit indices in structural equation modeling. Psychological Methods, 8, 16–37

Furthermore, authors are required to justify where they get the values to interpret the goodness of fit statistics, as well as to mention which software and version of them they used to conduct the analyses. These statistics are heavily contingent on a set of cut-off criteria.

Results

Authors need to justify why they use a 0.3 criterion to remove items, and not also others (e.g., cross loadings between factors), as well as to provide a reference that supports it.

Figure 1. Decimals on the Y axis are unnecessary. Authors are encourage to simplify the figure regarding the 58 potential numbers of factors, since the eigenvalue of 50 of them fall below 1.

Table 1. authors could simple not show factor loadings below a certain point to ease interpretation for the reader.

Authors need to provide an interpretation and its reference for the values of Reliability. Also, if the authors are to use Alpha as a way to speak of a questionnaire’s reliability, they need to speak about its assumptions. Otherwise, they need to use a different estimator (e.g., Omega).

Line 317 = it requires a reference.

Line 326 = once again, authors need to reference to support their criterion of 0.3 for factor loading

Once again, it is puzzling not to see the TLI estimator for the CFA in the second study. More so, I re-iterate the need for standardized and validated interpretations for the goodness of fit parameters.

Table 3. should have the name of the scales used.

Discussion

Authors should write an overall statement that reminds the readers of main topic and the importance of their research. Perhaps the first sentence could be moved in the paragraph, since the 2nd and 3rd sentences could do the job.

Line 438 = it requires a reference. In fact, the whole paragraph is discussed without providing a single reference.

This is actually repeated in a few other paragraphs in the discussion, which it is to a great extent unacceptable. Authors are required to discuss their findings at the light of previous ones. Overall, there are less than 10 items cited across the whole discussion.

Authors do not discuss the limitation of measuring attachment amongst adults only in romantic settings. Similarly, the limitations of the scale regarding its binary perspective are also not discussed. Considering this suggestion as a way to integrate my comment on the lack of references in the discussion section. Similarly, this should be done for every dimension of your instrument. Not every findings needs to be discussed, but all dimensions should.

none

Author Response

Please find below our responses in a point-by-point fashion (in italics). All changes in the main text are highlighted in green.

Congratulations on your submission. After reviewing your manuscript, my comments are mainly limited to a few considerations on the methods, and some things I believe missing. Otherwise, the rest of the comments are minor changes and suggestions I believe the authors will be able to handle without any problem.

R - We want to thank the reviewer for their appreciation on the paper and for the thoughtful and constructive comments. We revised the manuscript accordingly their suggestions. We believe that their comments helped us to significantly improve the paper. Our responses are reported below.

I have two major concerns: the absence of the TLI estimator in the analyses, and the lack of literature discussed in the discussion section. The former I explain more below; however, it is concerning that the estimator is not reported. The latter is simply unacceptable, and denotes the manuscript requires major changes in this section. The study results should not be offered by themselves. They co-exist within the literature. That is why the section is called that way, to enhance interpretability of the results and their impact. Below you may find these comments and suggestions, broken down by sections.

R - Below, we reply in detail to the reviewer’s comments. Overall, we have reported the TLI index and fully re-organized the discussion section to align with the more recent contributions in the literature.

Abstract

Line 17 = “fill this gap” sounds like the study does it all. “Contribute to fill the gap” seems more appropriate.

R - We thank the reviewer for this suggestion. We have changed the text accordingly.

Introduction

The first two paragraphs discus classical, yet very old literature on the topic. It is up the authors to consider removing them, for it they are not wrong. It just does not add anything to the article if taken out. Besides, it is a validation study, going that far in the theory is not justified. Of course, if taken out, the new first paragraph would need an appropriate new first sentence.

R - We have substantially modified the paragraphs in the introduction, which now begins with a more straightforward sentence that clearly justifies the purpose of the validation study. Additionally, we have reduced the mention of older theories and updated the references from the literature accordingly.

Line 57 = I believe the authors refer to “innate” rather than “inmate” rules.

R - We thank the reviewer for noticing this typo.

Methods

It is weird that I do not see the Tucker–Lewis index (TLI) index. This is perhaps a must in exploratory factor analysis. TLI evaluates the incremental improvement in fit of a given substantive model over that of a null model. Moreover, this statistic is often given along with the CFI.

Widaman, K. F., & Thompson, J. S. (2003). On specifying the null model for incremental fit indices in structural equation modeling. Psychological Methods, 8, 16–37

R - Done

Furthermore, authors are required to justify where they get the values to interpret the goodness of fit statistics, as well as to mention which software and version of them they used to conduct the analyses. These statistics are heavily contingent on a set of cut-off criteria.

R - We have now included the reference to the cut-off criteria and explicitly mentioned the software and its version employed in performing the analyses.

 Results

Authors need to justify why they use a 0.3 criterion to remove items, and not also others (e.g., cross loadings between factors), as well as to provide a reference that supports it.

R - We have now included the reference for the mentioned cut-off criteria.

Figure 1. Decimals on the Y axis are unnecessary. Authors are encourage to simplify the figure regarding the 58 potential numbers of factors, since the eigenvalue of 50 of them fall below 1.

R - We have simplified the figure according to the reviewer’s suggestions.

Table 1. authors could simple not show factor loadings below a certain point to ease interpretation for the reader.

R - We understand the reviewer’s point. However, it is generally recommended that all factor loadings are reported to ensure sufficient information for a full evaluation of the results. We hope the reviewer will agree with our decision to maintain the table as it is.

Authors need to provide an interpretation and its reference for the values of Reliability. Also, if the authors are to use Alpha as a way to speak of a questionnaire’s reliability, they need to speak about its assumptions. Otherwise, they need to use a different estimator (e.g., Omega).

R - We would like to respectfully point out that both coefficients (Cronbach’s alphas and McDonalds’ omegas) are reported in the manuscripts. We have also included an additional sentence in the text to report the results concerning the reliability coefficient adequacy.

Line 317 = it requires a reference.

R - Done

Line 326 = once again, authors need to reference to support their criterion of 0.3 for factor loading

R - Done

Once again, it is puzzling not to see the TLI estimator for the CFA in the second study. More so, I re-iterate the need for standardized and validated interpretations for the goodness of fit parameters.

R - Done

Table 3. should have the name of the scales used.

R - Unfortunately, this point was not addressed in the manuscript as we were uncertain about what the reviewer referred to. However, it's worth noting that Table 3 already includes the names of the scales utilized.

Discussion

Authors should write an overall statement that reminds the readers of main topic and the importance of their research. Perhaps the first sentence could be moved in the paragraph, since the 2nd and 3rd sentences could do the job.

R - We have now fully re-organized the discussion. We have strengthened the aim of the paper which was to validate a measure but at the same time we discussed the operationalized model in light of the contemporary debate on motivational systems.

Line 438 = it requires a reference. In fact, the whole paragraph is discussed without providing a single reference. This is actually repeated in a few other paragraphs in the discussion, which it is to a great extent unacceptable. Authors are required to discuss their findings at the light of previous ones. Overall, there are less than 10 items cited across the whole discussion.

R - Done. We have inserted here, and in other relevant points, new references.

Authors do not discuss the limitation of measuring attachment amongst adults only in romantic settings. Similarly, the limitations of the scale regarding its binary perspective are also not discussed. Considering this suggestion as a way to integrate my comment on the lack of references in the discussion section. Similarly, this should be done for every dimension of your instrument. Not every findings needs to be discussed, but all dimensions should.

R - We have modified the discussion section according to the reviewer suggestion. We have widened the section of the “limitations” adding other critical points, including the choice to limit the investigation to romantic love. Furthermore, we have included comments for each dimension. In the broader context, our discussion of the results has been framed within the ongoing contemporary discourse surrounding the connection between models of motivational systems and theories of personality traits.

Reviewer 2 Report

INTRODUCTION

The authors have provided an adequate overview of the importance of studying interpersonal motivations. However, they could strengthen their argument for the need for this new measure by more thoroughly reviewing the limitations of existing instruments. In addition, some evidence of associations between interpersonal motives and personality traits would be important.

This comment stems from the fact that the authors hypothesise that there are specific and numerous associations between the two constructs mentioned above.

ANALYTIC PLAN

Could you please provide the skewness and kurtosis cut-off values that you considered, with an accompanying literature citation?

Why do the authors not examine the divergent validity of the instrument? That is, is it perhaps possible to examine the degree of disagreement between indicators measuring different constructs?

The authors should present the reduced information on the characteristics of the participants (e.g. the omission of marital status) within the limits of the article.

There should also be some mention of the fact that only one sample was used for two different statistical analyses (this is a reduction in validity). Usually it is necessary to use two different samples for different studies and analyses for obvious reasons.

Considering that the sample is relatively large and that it is evenly distributed with regard to the biological sex. The test of instrument invariance could be considered by the authors. This would provide clinicians and researchers with a questionnaire that is valid for both sexes. There is no such indication from the generalised analysis considering the mixed sample.

The same consideration regarding age: the sample consists of persons between 18 and 80 years old, it might again be interesting to test the invariance for age to see if the instrument is valid for different age groups.

The section on convergent validity (as reported in a previous comment) seems to be a list of various correlations which in my opinion could be rewritten in a more schematic manner, but especially in the introductory part information on personality traits and interpersonal motivational systems must be provided.

DISCUSSION

The discussion is well done, but based on the above considerations, I would expand the limits section and, based on future corrections, add in the discussion section a consideration about the associations with personality traits, values and motivational systems.

Author Response

Please find below our responses in a point-by-point fashion (in italics). All changes in the main text are highlighted in green.

INTRODUCTION

The authors have provided an adequate overview of the importance of studying interpersonal motivations. However, they could strengthen their argument for the need for this new measure by more thoroughly reviewing the limitations of existing instruments. In addition, some evidence of associations between interpersonal motives and personality traits would be important.

This comment stems from the fact that the authors hypothesise that there are specific and numerous associations between the two constructs mentioned above.

R - We want to thank the reviewer for their positive feedback on the paper and for the thoughtful and constructive comments provided. We have revised the manuscript accordingly their suggestions. We believe that their comments helped us to significantly improve the paper. 

Overall, in the introduction, we have expanded on the significance of exploring motivational systems within the framework of personality trait theories. Specifically, we have introduced the concept by illustrating how motivational systems form an essential part of one's personality. As regards the instruments, we have emphasized the scarcity of available tools in the existing literature, thus underscoring the rationale behind our contribution.

ANALYTIC PLAN

Could you please provide the skewness and kurtosis cut-off values that you considered, with an accompanying literature citation?

R - We have now included this information.

Why do the authors not examine the divergent validity of the instrument? That is, is it perhaps possible to examine the degree of disagreement between indicators measuring different constructs?

R - We acknowledge that additional analyses might be conducted. To maintain clarity, we had to make specific choices. However, we address these aspects in the study's limitations section and also elaborate on them in the context of future research perspectives.

The authors should present the reduced information on the characteristics of the participants (e.g. the omission of marital status) within the limits of the article.

R - We have added this point in the study’s limitations.

There should also be some mention of the fact that only one sample was used for two different statistical analyses (this is a reduction in validity). Usually it is necessary to use two different samples for different studies and analyses for obvious reasons.

R - We would respectfully point out that two samples have been used in the present study, one for the EFA and the other one for CFA. Perhaps, the reviewer refers to the use of one sample for both CFA and convergent validity. However, it is generally accepted to have one sample for both CFA and the analysis of associations with convergent measures.

Considering that the sample is relatively large and that it is evenly distributed with regard to the biological sex. The test of instrument invariance could be considered by the authors. This would provide clinicians and researchers with a questionnaire that is valid for both sexes. There is no such indication from the generalised analysis considering the mixed sample. The same consideration regarding age: the sample consists of persons between 18 and 80 years old, it might again be interesting to test the invariance for age to see if the instrument is valid for different age groups.

R - We thank the reviewer for raising this point. However, we believe that measurement invariance would require a different sample of participants to ensure validity. We have mentioned this aspect as a future direction for research.

The section on convergent validity (as reported in a previous comment) seems to be a list of various correlations which in my opinion could be rewritten in a more schematic manner, but especially in the introductory part information on personality traits and interpersonal motivational systems must be provided.

R - As mentioned earlier, we have highlighted the link with personality traits by examining instances where there was a relationship between motivational systems and traits. Moreover, we have presented a clearer outline of the correlations between motivational systems and convergent measures.

Also, reporting correlations among measures as we did in the manuscript is a conventional approach generally used in the context of validation studies. We hope the reviewer will agree.

DISCUSSION

The discussion is well done, but based on the above considerations, I would expand the limits section and, based on future corrections, add in the discussion section a consideration about the associations with personality traits, values and motivational systems.

R - We are grateful for the reviewer's positive feedback. Nonetheless, we have extended this section by incorporating more recent references and framing the results within the ongoing contemporary discourse concerning the nature and functionality of motivational systems.

Round 2

Reviewer 1 Report

The authors properly addressed every point, justifying also their position on certain points brought up by this reviewer. Overall, I believe the quality of the manuscript has improved, while also complementing with the missing information, and correcting minor mistakes.

Given now that I am able to see the TLI scores, I believe the manuscript should include a discussion of the TLI scores as part of the discussion and limitation of the findings regarding the parsimony of the model. These values fall below current standards (TLI; ≥ .90 adequate; ≥ .95 good; Browne & Cudeck, 1993; Marsh et al., 2005; Schermelleh-Engel et al., 2003), perhaps considering the degrees of freedom of the model tested.

Marsh, H. W., Hau, K.-T., & Grayson, D. (2005). Goodness of fit evaluation in structural equation modeling. In A. Maydeu-Olivares & J. McArdle (Eds.), Contemporary Psychometrics (pp. 275–340). Erlbaum

- Browne, M. W., & Cudeck, R. (1993). Alternative ways of assessing model fit. Testing Structural Equation Models, 21(2), 136–162. https://doi.org/10.1167/iovs.04-1279

Schermelleh-Engel, K., Moosbrugger, H., & Müller, H. (2003). Evaluating the fit of structural equation models: Tests of significance and descriptive goodness-of-fit measures. MPROnline, 8(2), 23–74 

Author Response

The authors properly addressed every point, justifying also their position on certain points brought up by this reviewer. Overall, I believe the quality of the manuscript has improved, while also complementing with the missing information, and correcting minor mistakes.

Given now that I am able to see the TLI scores, I believe the manuscript should include a discussion of the TLI scores as part of the discussion and limitation of the findings regarding the parsimony of the model. These values fall below current standards (TLI; ≥ .90 adequate; ≥ .95 good; Browne & Cudeck, 1993; Marsh et al., 2005; Schermelleh-Engel et al., 2003), perhaps considering the degrees of freedom of the model tested.

- Marsh, H. W., Hau, K.-T., & Grayson, D. (2005). Goodness of fit evaluation in structural equation modeling. In A. Maydeu-Olivares & J. McArdle (Eds.), Contemporary Psychometrics (pp. 275–340). Erlbaum

- Browne, M. W., & Cudeck, R. (1993). Alternative ways of assessing model fit. Testing Structural Equation Models21(2), 136–162. https://doi.org/10.1167/iovs.04-1279
Schermelleh-Engel, K., Moosbrugger, H., & Müller, H. (2003). Evaluating the fit of structural equation models: Tests of significance and descriptive goodness-of-fit measures. MPROnline8(2), 23–74 

R - We are pleased that our efforts in addressing the points raised in the first revision round and our justifications have been well-received. Also, we greatly appreciate the reviewer's further insightful suggestions regarding the use of the Tucker-Lewis Index (TLI).

In response to the reviewer's suggestion to discuss TLI scores as part of the discussion and limitation of the findings regarding model parsimony, we would like to respectfully highlight that the TLI values for the most optimal models we identified are above the standard threshold criteria. Indeed, as underscored by the reviewer, TLI scores are conventionally evaluated against established benchmarks where values of ≥ 0.90 indicate acceptable fit and ≥ 0.95 suggest robust fit (Browne & Cudeck, 1993; Marsh et al., 2005; Schermelleh-Engel et al., 2003).

Acknowledging this, we reported that in our preliminary exploratory factor analysis, the TLI stood at 0.89, which is slightly below the suggested threshold.

Throughout the process of refining our model, we adopted a systematic approach by identifying and subsequently excluding items with factor loading values below 0.30. This iterative procedure yielded enhancements in the model fit, evidenced by the increased TLI score of 0.902 in the final EFA, alongside other aspects of model evaluation that contribute to ensuring the acceptability and utility of the solution. [page 5, lines 246-247].

In the context of confirmatory factor analysis, we maintained the same approach, comprehensively evaluating the model's fit, overall interpretability, size, and statistical significance of the model parameters.

More specifically, in the analytic approach section, we specified that:

“To evaluate measurement goodness, we used the cut-off of 0.30 for the inspection of standardized factor loadings (i.e., items that had factor loadings less than 0.30 on their target factor or greater than 0.30 on non-target factors were excluded), and the range of 0.10-0.90 for acceptable residual error variances (i.e., items with residual variances less than 0.10 or greater than 0.90 were removed)”. [page 12, lines 334-343].

As reported in the results section of the manuscript, “the U-CFA performed on the 58 items resulting from the EFA in Study I presented an adequate model fit, χ2 = 2132.230, df = 1217; CFI = 0.920; RMSEA = 0.034 [0.032, 0.037]; SRMR = 0.027; BIC = 81594.527, except for TLI, which slightly fell below the established threshold (TLI = 0.89). Also, when looking at the parameter estimates, the solution showed that four items (see items 6, 20, 21, and 22 in Table 1) had loadings lower than 0.30 on their target factor. Three of these items also presented significant loadings higher than 0.30 on non-target factors (6, 21, and 22). Furthermore, four items presented cross-loadings exceeding 0.30 (see items 12, 23, 33, and 56 in Table 1).” [page 13, lines 356-363]

As a result of the refinement process based on the aforementioned criteria, the model without the critical items “showed an overall improvement in model fit, as indicated by the lower BIC value (BIC = 70867.820; overall model fit: χ2 = 1499.060; df = 853; CFI = 0.932; TLI = 0,905; RMSEA = 0.035 [0.032, 0.037]; SRMR = 0.025). All items presented factor loadings higher than 0.30, and no critical cross-loadings were detected. Residual variances ranged from 0.371 (item 24) to 0.746 (item 2).” [page 13, lines 363-369].

In the subsequent phase, to ensure model parsimony and reduce complexity, we used the Wald test to identify cross-loadings that could be feasibly constrained to zero without compromising the model's fit. The results are reported below:

“As a second step, we moved to inspect nonsignificant cross-loadings that could be constrained to zero. Two hundred ninety-six loadings were nonsignificant and then constrained to zero. These changes did not result in a decline in fit, χ2 = 1842.7565, df = 1082; CFI = 0.922; TLI = 0.912; RMSEA = 0.033 90% C.I. [0.031, 0.036], SRMR = 0.036, BIC = 69680.579, ΔRMSEA = 0.002 and ΔCFI = 0.010”. [page 13, lines 370-375].

Finally, we compared the partially unrestricted and fully constrained CFA models, finding that the U-CFA model, despite having more free parameters, demonstrated a more satisfactory fit to the data than the traditional CFA model.

“Finally, we ran a traditional CFA and compared it to the factor structure resulting from the final U-CFA using the criteria introduced earlier. Overall, the CFA model does not achieve a reasonable fit by conventional standards, χ2 = 2389.191, df = 1147; CFI = 0.873; TLI = 0.864; RMSEA = 0.041 90% C.I. [0.039, 0.044], SRMR = 0.054, BIC = 69869.585, and the change in CFI exceeds the frequently used benchmarks of ΔCFI ≤ 0.010”. [page 13, lines 377-381].

To ensure clarity, we have now specified that this result supports the notion that permitting the identified cross-loadings within the U-CFA model better captures the underlying structure of the data.

“These findings indicated that the U-CFA model, despite having more free parameters, demonstrated a more satisfactory fit to the data than the traditional CFA model. This supports the notion that permitting the identified cross-loadings within the U-CFA model better captured the underlying structure of the data”. [page 13, lines 381-385].

While we are uncertain about the specific limitation the reviewer is suggesting about discussing TLI and overall model parsimony, we hope that the clarifications provided in this revised version of the manuscript address the reviewer's concerns. We remain open to further elaboration on this matter if needed and value the reviewer's feedback as it contributes to enhancing the clarity and depth of our study.

Reviewer 2 Report

As far as I am concerned, the authors did a good job of clarifying the issues at stake. 

Perform a final proofreading of your manuscript. Reading your work aloud can help you catch errors that might have been overlooked in earlier readings.

Author Response

As far as I am concerned, the authors did a good job of clarifying the issues at stake. 

Perform a final proofreading of your manuscript. Reading your work aloud can help you catch errors that might have been overlooked in earlier readings.

R – We thank the reviewer for their feedback and valuable guidance in improving the overall quality of the manuscript.

Following the reviewer’s suggestion, we have carefully proofread the manuscript, ensuring its overall accuracy and clarity.

Round 3

Reviewer 1 Report

Authors address all concerns raised satisfactorily.